# The Role of eHsp90 in Extracellular Matrix Remodeling, Tumor Invasiveness, and Metastasis

**DOI:** 10.3390/cancers16223873

**Published:** 2024-11-19

**Authors:** Pragya Singh, Daniel G. Jay

**Affiliations:** Graduate School of Biomedical Sciences, Department of Developmental, Molecular and Chemical Biology, Tufts University School of Medicine, Boston, MA 02111, USA

**Keywords:** cancer invasion, metastasis, tumor microenvironment, extracellular matrix, basement membrane, interstitial matrix, matrix metallo proteinase, Collagen-1, extracellular Hsp90

## Abstract

Hsp90 is a protein that is often found in high levels inside cancer cells, helping them grow and spread. Yet, it is difficult to target with treatments because it’s involved in many important normal cell functions. However, a subset of this population found outside the cells specifically cancer cells, called extracellular Heat-shock protein 90 (eHsp90), could be a better target for cancer therapy. eHsp90 plays a key role in helping cancer spread systemically by affecting the environment around the tumor, especially the extracellular matrix (ECM). eHsp90 changes the amount, nature and architecture of the ECM that surrounds the tumor and helps it spread locally within the tissue, which is a crucial step before ultimately metastasizing to other tissues as shown in mice models. Thus, targeting eHsp90 could help stop tumors from spreading by disrupting these processes.

## 1. Introduction

Since the discovery of an extracellular population Hsp90 (eHsp90), it has been implicated in the several pathophysiological conditions including wound healing and fibrosis, and especially cancer [1]. eHsp90 offers a potentially safer and more effective therapeutic strategy compared to pan-Hsp90 inhibitors, which affect intracellular Hsp90 and its many homeostatic functions such a correct protein folding during synthesis; activate proteins to regulate their function; facilitate ligand binding; and transcriptional regulation. Hsp90 functions reviewed in [2,3,4]. Since Hsp90 is critical for cell homeostasis and interacts with approximately 200 proteins, inhibitors developed to target the intracellular Hs90’s ATP-dependent chaperone function has encountered setbacks in cancer clinical trials due to high toxicity observed at effective doses likely due to interference with Hsp90’s function of homeostasis [5,6]. There have been 18 small-molecule Hsp90 inhibitors into cancer clinical trials since 1999, none have yet received FDA approval [7]. eHsp90 on the other hand is a promising therapeutic target, given its involvement and secretion in minimal physiological processes like protection against proteotoxic, oxidative and ischemic stresses and wound healing. Reviewed in detail by [1,8].

Majority of eHsp90 functions have been studied in cancer particularly in tumor invasion, which represents the first step in the metastasis cascade [1,9]. Metastasis remains the leading cause of cancer related mortality, with mortality rates dropping to 30% survival rate beyond 5 years once the cancer has metastasized [10,11]. This underscores the urgent need to better understand metastasis and its key drivers to develop targeted therapies. Metastasis involves a complex series of events known as the “invasion-metastasis” cascade in which a tumor spreads from its site of origin to secondary organs and tissues [12]. First, in a two-part invasion step, tumor cells first breach the boundary created by the basement membrane (BM), followed by infiltration and traversing the connective tissue stroma comprised of interstitial matrix (IM), and mesenchymal cells [13,14]. In the next step, invading cells upon locating lymphatic or blood vessels in the stroma, enter the circulatory system as circulating tumor cells [14,15]. These cells then disseminate systemically through the blood and lymphatic vessels, eventually undergoing extravasation and colonizing secondary sites, forming microscopic lesions that may progress into macroscopic metastatic lesions [16,17]. Several efforts are underway to understand the metastatic cascade with the goal of disrupting it at various stages. Among these stages, invasion stands out as a critical point for intervention, as targeting it offers the most effective means of inhibiting the cascade while the cancer remains confined to a single location.

eHsp90 as a therapeutic target is an exciting new avenue to explore to limit the tumor invasion and metastasis. A large body of research collectively shows that eHsp90 may serve as a hub interacting with a cohort of proteins outside cancer cells to enhancing both invasion and metastatic potential [18,19,20,21,22,23,24,25,26]. Inhibiting eHsp90 could thus offer therapeutic benefits for treating metastatic cancers. Studies have shown that eHsp90 promotes invasiveness across various cancers, initially in fibrosarcoma and subsequently in breast, melanoma, colorectal, prostate, Bladder lung cancers, and glioblastoma [18,19,20,21,22,23,25,26,27,28,29]. Inhibiting eHsp90 using various inhibitors and Abs developed to target the extracellular form without affecting intracellular Hsp90 not only inhibits in-vitro invasion, in animal models, these inhibitors have demonstrated benefits, including reduced metastatic lesions and improved survival rates [18,19,20,21,22,23,24,25,26,27,28,30,31,32,33,34,35,36,37,38,39,40,41,42,43]. One way that eHsp90 augments tumor invasiveness by upregulating signaling pathways such as HER-2, and LRP1 in cancer cells [19,23,41]. However, eHsp90 is primarily known for enhancing the invasiveness of cancer cells by interacting with ECM proteins and ECM-modifying proteins present in the TME, such as Matrix Metalloproteinase (MMP)-2, MMP9, Plasminogen, Collagen-1, Fibronectin (FN), and LOX-like protein-2 (LOXL2) [18,20,21,22,24,25,26,27,44,45,46].

Given the abundance of ECM in the TME and its pivotal role in tumor malignancy, particularly in invasion and metastasis, targeting the ECM holds promise as a therapeutic strategy to limit metastatic spread [47]. Understanding how eHsp90 affect the ECM would aid in optimizing this strategy. This review discusses the role of the ECM in invasion and metastasis, highlighting hallmark alterations in the ECM matrix observed in invasive tumors. Furthermore, it explores the role of eHsp90 as a master regulator of ECM proteins and proteases crucial in invasion, underscoring its potential as a target for developing anti-invasion and metastasis therapies

## 2. Structure and Composition of the ECM in Tissues and Its Deregulation Within the TME

The ECM is a dynamic network comprising approximately 300 macro proteins, categorized into fibrous proteins, glycoproteins, and proteoglycans, crucial for tissue structure and function. In healthy tissues, the ECM serves multifaceted roles of providing structural support and regulating biochemical signals essential for tissue homeostasis [48]. Structurally, the ECM proteins assemble in two primary forms: the BM and the IM. The BM, consisting of Collagen-IV and Laminin, forms a dense, two-dimensional sheet-like structure, which separates the epithelial and/or endothelial compartment from the stromal compartment of the tissue [49]. The IM in contrast is primarily composed of Collagen-1 fibers, interspersed with FN and Elastin at intersections of Collagen-1 fibers, is found in the stromal compartment and surrounding the mesenchymal cells within tissues, providing scaffold for tissue organization [48,50] (See Figure 1).

In healthy and functional tissues, the ECM composition, density, stiffness and structure are precisely controlled to ensure tissue integrity, function and homeostasis [52,53,54]. However, in ECM surrounding the invasive tumors, the tight regulation is lost, and abnormalities in the ECM becomes a hallmark characteristic of the TME [55]. The invasive tumors exhibit progressive ECM abnormalities such as deregulated ECM deposition and degradation, altered structure and orientation of the ECM fibers, and changes in ECM mechanical properties such as increased stiffness [47,56,57]. Cancer cells traverse the ECM during the invasion to find an escape route, i.e., a circulatory vessel [58]. Thus, the altered ECM and its cues to cancer cells are relevant in cancer invasiveness and metastasis [55,56,57,58] (See Figure 1).

These pathological alterations in the ECM surrounding invasive tumors are unsurprisingly facilitated by tumor cells themselves, as they upregulate the expression, activity, and stability of ECM proteins, ECM-regulating enzymes and fibroblast activation in the TME to promote the malignant spread, by secreting proteins and cytokines that facilitate these processes—such as eHsp90 [59,60,61,62]. eHsp90 secreted by cancer cells, acts as a master regulator of several ECM components and ECM-regulating enzymes, potentially influencing overall ECM dynamics within the TME [18,20,21,22,24,26,27,29,44,45,46]. This pivotal role allows eHsp90 to possibly exert control over ECM prevalence, composition, and structure, discussed below. Understanding the intricate and multifaceted interplay between eHsp90 and ECM remodeling provides valuable insights into its potential as both a prognostic marker and therapeutic target for managing tumor invasion and metastasis in human cancers.

## 3. Cancer Secreted Hsp90’s Role in Invasion and Metastasis as an ECM Modulator

In this section, we discuss eHsp90’s interactions with various ECM proteins, ECM-modifying enzymes, and cells within the TME, highlighting its role in altering ECM degradation, deposition, structure, and dynamics, thereby promoting tumor invasion and subsequent metastasis (Summarized in Table 1).

### 3.1. eHsp90 Activates and Stabilizes ECM Proteases

One of the key roles of eHsp90 is orchestrating the activation and regulation of ECM proteases which are known to promote invasion and metastasis, particularly MMPs and Plasminogen [18,20,21,22,24,45].

MMPs, a diverse family of 23 proteases, play a vital role in degrading ECM proteins and facilitating cancer cell invasion [64,65]. Proper regulation of MMPs is essential for maintaining the ECM integrity and tissue function [64,65]. Cancer cells disrupt this delicate balance through various mechanisms, including upregulating MMP expression and activation, as well as dysregulating compartmentalization, and degradation [66,67,68]. Abnormally high MMP activity results in the enhanced degradation of ECM components, particularly the [67,68]. MMP-led degradation creates breaches in the sheet-like structure of the BM, enabling cancer cells to escape the confines of the epithelial compartment and invade surrounding tissues [67,69]. Beyond ECM degradation, MMPs also influence other processes within the TME, including cell-signaling, apoptosis, immunomodulation, and neovascularization, all pivotal for tumor progression, invasion and metastasis [64,65,66]. Understanding the factors that facilitate cancer cells in dysregulating MMPs therefore can be the key to inhibit the problematic and abnormal function of MMPs and thus tumor invasion.

eHsp90 plays a crucial role as a regulator of MMPs, particularly MMP-2 and MMP-9, by promoting their activation and stability in the extracellular environment [18,21,22,24,43,45]. Our laboratory was the first to demonstrate the interaction between MMP-2 and eHsp90 [18]. Subsequent studies corroborated that eHsp90 interacts with cancer cell secreted MMPs- particularly the inactive zymogen forms of MMP-2 and MMP-9 [21,22,24,42,43,45]. Through these interactions, eHsp90 activates MMP-2 and MMP-9, converting them into their functionally active forms [18,21,22,24,42,43,45]. This activation significantly enhances their protease activity, which in turn promotes the invasiveness of cancer cells, including those in fibrosarcoma and breast cancer cells [18,21,24,42,43].

Another key mechanism through which eHsp90 regulates MMP activity involves stabilizing their active forms, preventing active MMP degradation. Building on its role in activating MMPs, eHsp90 was shown in a study by Baker-Williams et al. to interact directly with active MMP-2 and utilizing co-chaperones such as AHA1 and TIMP2, eHsp90 stabilizes the active form of MMP-2, shielding it from further proteolysis and sustaining its prolonged proteolytic activity [45].

To regulate the activation and stabilization of MMPs, eHsp90 recruits assistance of various co-chaperones and docking proteins. A study from our lab by Sims et al. showed that co-chaperones, such as AHA1, P23, and Hop, which are known to associate with intracellular Hsp90, interact with and assist eHsp90 in activating MMP-2 [24]. Corroborating this, another study by Baker-Williams also showed that eHsp90 excised control over MMP-2 activation and stabilization depends on the specific co-chaperone it binds [45]. The authors showed that binding of an activating co-chaperone like AHA1 enhanced eHsp90-led MMP-2 activation and stabilization, whereas binding of an inhibitory co-chaperone like TIMP-2 led to a reduced active MMP-2 [45].

Other than co-chaperones, post-translational modifications of eHsp90 itself are crucial in dictating its ability to bind and activate MMPs. For instance, hyperacetylation of eHsp90 achieved by inhibiting deacetylase in cancer cell condition-media led to an increase in MMP-2 binding and activation [42]. Overall, eHsp90- mediated regulation of MMP activity is complex and should be explored further for developing effective therapeutic strategies.

In addition to regulating MMPs, eHsp90 also activates another extracellular protease, tissue plasminogen activator (tPA) [20]. McCready et.al. identified tPA bound to eHsp90 in the media of fibrosarcoma and breast cancer cells [20]. tPA converts the zymogen plasminogen into the active protease plasmin, and inhibiting eHsp90 in this study led to decreased active plasmin levels, which ultimately reduced cancer cell migration [20]. Similar to MMPs, Plasmin degrades various ECM proteins such as FN, Laminin, and Tenascin-C [70,71]. It also has other functions including redistribution of growth factors, promoting protumorigenic processes such as inflammation, angiogenesis and activation of other ECM proteases including MMPs [70,71]. The direct impact of eHsp90 inhibition on all active Plasmin functions has not been studied. However, it is highly probable that inhibiting the eHsp90-tPA interaction would block the premetastatic functions of deregulated Plasmin.

While the studies mentioned above highlight eHsp90’s role in activating the ECM proteolytic functions of MMPs and tPA, which enhance cancer cell migration and invasion through the BM, these proteins also influence other pro-metastatic processes such as intravasation, extravasation, and colonization [42,45,64,65,66,70,71,72]. This underscores the importance of inhibiting eHsp90 activity to potentially block other stages of metastasis as well. Furthermore, research into how eHsp90-mediated activation of MMPs and tPA affects overall changes in ECM characteristics within the TME could deepen our understanding of eHsp90’s impact on the TME and provide valuable insights for developing targeted therapeutic strategies to disrupt these processes and limit metastatic spread.

### 3.2. eHsp90’s Role in ECM Deposition

The increased deposition of ECM within the TME often makes up a substantial portion of an advanced tumor mass and is an established hallmark of invasive cancers [56,57]. The excessive buildup of ECM actively facilitates the malignant behavior of cancer cells by being conducive to their dissemination [57,73,74,75]. Specifically, the accumulation of ECM molecules such as Collagen-1 and FN enhances cancer cell migration and invasion by promoting directional cell movement and invadopodia formation [73,76]. Higher ECM concentrations are also linked to increased migration speed and overall collective cancer cell invasion [73]. Beyond a direct effect, the mechanical changes caused by ECM deposition such as increased tissue stiffness and the resulting tissue stress—further contribute to the tumor malignancy [77]. Infact, increased tissue stiffness resulting from ECM deposition is a recognized driver of cancer invasion and an independent indicator of poor prognosis, including higher rates of invasion and mortality [78]. The stiffness of the tumor stroma (~400 Pa) is significantly higher than healthy stroma (150 Pa), and this difference has been identified as an independent prognostic marker for metastasis [78,79].

This surplus in deposition of ECM in the TME can stem from either increased production and secretion of the ECM molecules by cells or a reduced turnover and reinternalization of the ECM. eHsp90 plays a key role in both processes—likely promoting the secretion of ECM components and inhibiting their reinternalization, there by contributing to the excessive accumulation of ECM molecules within the TME [25,44]. This function of eHsp90 is exemplified by its regulation of FN levels in the extracellular environment of cancer cells [25,44]. Hunter et al. showed that adding exogenous Hsp90 to the conditioned media of breast cancer cells resulted in increased FN deposition, while inhibition of Hsp90 using a C-terminal inhibitor (Novobiocin) and siRNA reduced the FN levels in the ECM matrix [25]. The authors suggested that the elevated FN presence was due to eHsp90-mediated inhibition of FN reinternalization, allowing for its accumulation in the ECM. [25]. The mechanism by which eHsp90 regulates FN internalization was further elucidated by Boel et al., who showed that eHsp90 prevents the binding of FN to its receptor, LRP1, a known client protein of Hsp90, thus blocking its reinternalization [44]. FN is a critical component of the IM—responsible for stabilizing the IM network and activating integrin-mediated cell signaling pathways [80]. In cancer, dysregulation of FN commonly results in increased cell migration and invasion [81]. Thus, targeting eHsp90 to limit FN abundance could potentially prevent invasive processes and reduce metastasis [82].

Beyond its direct regulation of the ECM proteins, eHsp90 may also influences ECM deposition by activating stromal cells present in the TME. Within the TME, ECM secretion leading to matrix deposition and alteration is primarily carried out by resident fibroblasts that have transformed into myofibroblast-like cells found in healing wounds called Cancer “Activated” Fibroblasts (CAFs) [61,83,84]. CAFs, the largest population of stromal cells can make constitute up to 80% of TME in the advanced solid cancers [83,84]. An increased population of CAFs in the TME along with increased deposition of ECM leads to Desmoplasia—a phenomenon linked to cell reprogramming, metastasis, and resistance to therapy in many cancers [85,86]. In the TME, CAFs not only deposit the ECM molecules but actively regulate ECM architecture by secretion of factors and exerting mechanical force, making the TME more proinvasive [87]. The stromal expression of CAF markers, particularly Tenascin-C, has been shown to predict worse outcomes in several cancers [88]. Acting as a tumor’s paracrine messenger eHsp90 activates resident fibroblasts into CAFs. [89]. Acting as a paracrine messenger, eHsp90 activates resident fibroblasts into CAFs. For example, a prostate cancer study by Bohonowych et al. demonstrated that fibroblasts exposed to eHsp90 showed increased expression key CAF markers, including αSMA, vimentin, fibroblast activation factor, and Tenascin-C [90]. eHsp90-induced transformation of fibroblasts into CAFs enhanced cell motility and stimulated CAFs to secrete pro-inflammatory cytokines, such as IL-6 and IL-8, which are involved in cancer progression and invasion [90]. Although likely, a direct effect of eHsp90-activated fibroblasts on increased ECM deposition was not explored in this study and is an important area for further investigation. eHsp90’s role in activating fibroblasts is supported by other studies on cardiac, lung, and pulmonary fibrosis, which have shown that eHsp90 activates TGFβ signaling in fibroblasts, leading to their transformation into αSMA-expressing myofibroblasts [89,91,92].

### 3.3. eHsp90’s Role in Restructuring the ECM

In healthy tissue, the three-dimensional IM in the stromal compartment is organized as a mesh of thin, curly, randomly oriented amorphous fibers [93,94,95]. However, as tumors originate and progress, a noticeable linearization and bundling of ECM fibers occurs within the tumor microenvironment (TME) [94,96,97]. This structural change, which involves the straightening and bundling of fibers, is accompanied by a progressive radial orientation of the ECM fibers around the tumor’s periphery, particularly in invasive tumors [94,96,97]. In contrast to the unorganized, thin fiber mesh of healthy tissue, the remodeled IM fibers in the TME form track-like structures, particularly Collagen-1 and, to a lesser extent, FN. [94,96,97]. These bundled and peripherally aligned fibers provide a path of lower resistance, facilitating cancer cell migration and invasion beyond the tissue of origin [94,96,97]. In breast cancer, the progression of Collagen-1 morphology into anisotropic structures is a clinically recognized marker of the invasive tumor and is used to predict patient outcomes [94,95,98]. The progression of altered Collagen-1 morphology referred to as Tumor Associated Collagen Morphology (TACS) has been captured in mouse tumors and has been categorized into distinct stages that directly correlates with tumor progression and invasiveness [94,95,99]. TACS-1 is characterized by increased Collagen deposition density found surrounding a small, localized tumor, while TACS-2 is marked by the straightening and bundling of Collagen-1 fibers, which coincides with tumor growth in size and early invasion [94,95,99]. In TACS-3, Collagen-1 fibers further straighten and bundle together accompanied with fiber alignment with each other and radial orientation around the tumor—signaling extensive invasion beyond BM and infiltration deep into the IM [94,95,99]. The remodeled Collagen-1 fibers in TACS-3 are preferentially utilized by tumor cells as tracks to invade out of the confines of the primary tissue [94,95,99]. Additionally, mesenchymal stromal cells such as immune cells etc. also use these tracks to infiltrate the TME, thus changing the overall landscape of the TME [97]. Importantly, TACS has demonstrated prognostic value in various cancers, highlighting its significance in predicting patient outcomes [35,94,95,97,99,100,101].

The remodelling of Collagen-1 fiber and of other ECM molecules is known to be carried out by the Lysyl Oxidase (LOX) family of proteins including LOX and LOX-like proteins (LOXL) 1–4 [102]. A study by McReady et al. from our lab demonstrated that eHsp90 directly interacts with LOXL2 protein in the CM of the breast cancer cells [27]. LOXL2 is known to mediate crosslinking of Collagen-1 fibers resulting in increased stiffness of the TME, subsequently resulting in EMT, increase in cell motility, and invasiveness [103,104].

Our recent study conducted by Singh et al., identified a new interaction between Collagen-1 and eHsp90, while exploring the involvement of eHsp90 in Collagen-1 crosslinking and alignment via LOXL2 [26]. In this study we discovered that eHsp90 directly interacts with Collagen-1 and autonomously orchestrates fiber alignment, thereby promoting invasion of breast cancer cells [26]. Additionally, we explored the structural and functional mechanisms of Hsp90 that are crucial for its interaction with Collagen-1 and subsequent fiber alignment revealing that eHsp90’s binding to Collagen-1 is not contingent upon ATPase activity [26]. Instead, the open conformation of Hsp90’s N-terminal, assumed in the absence of Hsp90-ATP drives direct interaction with Collagen-1 and facilitates fiber alignment [26]. Furthermore, our study underscored the significance of Hsp90 dimerization in executing Collagen-1 alignment [26]. These findings provide crucial insights into the comprehensive impact and mechanism of eHsp90 on most abundant ECM molecule structure i.e., Collagen-1 and invasion of Cancer cells through the Collagen-1 matrix, which could suggest the development of anti-invasion therapeutic molecule targeting eHsp90.

Another IM molecule that eHsp90 directly interacts to alter the matrix structure is FN [25,44]. In a study conducted by Chakraborty et al., FN matrix deposited by cancer cells displayed thicker FN fibers when exposed to eHsp90 in the CM [46]. Here, the increase in the thickness of FN fibers can be attributed at least in part, to the bundling of FN fibers, as eHsp90 also has function in preventing the internalization of FN, thus increasing the overall abundance of the FN resulting in thicker fibers [46]. While these studies provided direct evidence that eHsp90 addition leads to thicker FN fibers, either through increased deposition or fiber bundling, a direct connection between increased FN matrix caused by eHsp90 and cancer cell invasion has yet to be established.

## 4. eHsp90 Enhances Invasion and In Vivo Metastasis

The pro-invasive role of eHsp90 observed in-vitro is corroborated by in-vivo studies demonstrating its crucial role in invasion and metastasis across various animal cancer models. (Summarized in Table 2) Numerous studies, including those by Becker et al. and Crowe et al., have demonstrated the presence of tumor cell surface-bound Hsp90, particularly in malignant tumor types, while Milani et al. reported a 10- to 15-fold increase in soluble Hsp90 in the plasma of mice grafted with primary human acute lymphoblastic leukemia compared to healthy controls [38,105,106].

While these studies indicate that eHsp90 is significantly associated with malignant and metastatic tumors, a direct role of secreted eHsp90 in driving the metastasis is shown in various mouse models of cancer [38,106]. For instance, Zou et al., showed that in contrast to parental MB231 breast cancer cells, eHsp90α Knockout (KO) MB231 cells, failed to form visible primary tumors and lung metastases [37]. Moreover, the tumorigenicity and metastatic potential of the MB231 cells with Hsp90α-KO was rescued by intravenously injecting in mice the recombinant wild-type Hsp90α [37]. Similarly, Gong et al. inhibited the secretion of Hsp90α using the inhibitor Metformin, resulting in decreased local invasion and metastasis of lung cancer cell line H1299 when they were orthotopically implanted [40]. In this study, parallel tests were performed by injecting recombinant Hsp90 (rHsp90α) alone which showed an increased local invasion and liver metastasis of H1299 tumors [39]. Plus, rHsp90α successfully reversed the inhibitory effects of Metformin in liver metastasis of H1299 lung tumors, clearly establishing the critical role of secreted Hsp90α’s in driving invasion and metastasis [40]. Another study by Hou et al., utilized rHsp90α intravenous injection and showed that in established non-invasive and metastatic mammary tumors formed by MCF-7 cells could be induced to metastasize to lymph nodes when exposed to rHsp90α, while saline injected mice were not able to form the metastatic nodes [39].

The critical role of eHsp90 in metastatic spread has been further solidified by numerous studies showcasing that direct inhibition of eHsp90 activity resulted in reduced metastatic spread [21,28,30,31,32,33,36,107]. For instance, Tsutsumi et al. showed that exposing the B16 melanoma cells to DMAG-N-oxide, a cell-impermeant N-terminal inhibitor of Hsp90, prior to injecting them into mice resulted in a reduced metastatic focus within the lungs, when compared to metastatic deposits in the mice injected with untreated B16 cells [32]. It is important to highlight that cell-impermeant eHsp90 inhibitors are anticipated to be significantly less toxic to normal, non-cancerous cells than cell-permeable pan-Hsp90 inhibitors. Consequently, these inhibitors will likely be better tolerated by patients, potentially leading to more favorable outcomes in clinical trials compared to cell-permeable inhibitors of Hsp90.

Similar reduction in metastasis has been observed when eHsp90 activity is inhibited using monoclonal antibodies (mAb) [21,30,31,33,37,39]. For instance, Patsavoudi lab developed an eHsp90 targeting mAb-4C5 and showed that it prevents the deposition of intravenously injected B16F10 melanoma cells into lungs [19,21,30,33,37]. A similar effect of reduction in lung metastasis of injected B16 melanoma cells is demonstrated by Wang et al. by treating the mice with a different eHsp90 neutralizing antibody post cell injection [31]. Wang et al. made a noteworthy observation in this study—mice treated with metastatic deposits in the mice treated with Hsp90α mAb exhibited clearly defined boundaries, contrasting the extensive stromal infiltration of metastatic foci observed in mice treated with IgG [31]. Although the invasion observed in the study is not in the primary tumor, it is possible that inhibition of eHsp90 likely influences invasion in the primary tissue in-vivo, and the local spread of processes into primary and secondary tissues. This group also utilized the eHsp90 neutralizing antibody and showed a stark reduction in the lymph node and liver metastasis in the orthotopic mouse model of breast cancer in comparison to IgG [31]. Zou et al. developed and used another monoclonal antibody (mAb) called IG6-D7, targeting the F-5 motif of Hsp90α that lies in the linker region between N and M domain to address eHsp90 in breast cancer [37]. When MB231 breast cancer cells were co-injected with IG6-D7, there was a significant reduction in the formation of metastatic deposits in nude mice compared to mice treated with IgG-MB231 [37]. These studies collectively establish that eHsp90 is crucial for in-vivo cancer invasion and metastasis, underscoring its potential as a biomarker and therapeutic target for anti-metastasis treatments in clinical settings. Moreover, the previously discussed eHsp90-binding inhibitors such as DMAG-N-oxide and IG6-D7 may be utilized to clinically target eHsp90 or inform next approach for anti-eHsp90 therapy to curb invasion and metastasis. Although more mechanistic information is required to determine the exact mechanism to inhibit eHsp90 for being most effective and safe at limiting metastasis, and the effect on eHsp90’s client proteins. This includes understanding the exact pathways through which eHsp90 promotes metastasis and the impact of its inhibition on eHsp90’s client proteins. These insights will help develop precise and effective clinical interventions to limit metastasis.

## 5. eHsp90 as a Marker for Invasive Cancer in Clinic

Multiple studies demonstrate that elevated levels of eHsp90 in the plasma and serum are associated with various invasive and metastatic cancers. Burgess et al. first demonstrated a significant increase in Hsp90 levels, at least two-fold, in the plasma of prostate cancer patients compared to the normal population [108]. Similarly, Wang et al. found elevated plasma Hsp90 levels in cancer patients relative to healthy individuals, establishing a positive association between plasma Hsp90 levels and malignancy and metastasis in breast, lung, pancreas, and liver cancers [31]. Notably, plasma Hsp90 levels were significantly higher in metastatic tumors compared to both healthy cohorts and benign tumor patient samples. Interestingly, in breast cancer, plasma Hsp90 levels did not correlate with primary tumor size or estrogen and progesterone receptor expression, suggesting a specific association with metastatic potential rather than tumor size [31]. Similarly, various large population cohort studies for Multiple myeloma [109], Lung [110], Liver [111], Melanoma [112], Colorectal [113], Gastric [114,115], Hepatic [116], and Breast cancer [39] have consistently shown that plasma or serum Hsp90 is elevated in patients with invasive and metastatic cancers compared to healthy individuals and patients with benign tumors. These studies have concluded that secreted Hsp90 is a potent biomarker for these cancers. In a study Liu et al., Hsp90 levels in the plasma of the patients er able to predict the presence of different cancers, including Breast, Lung, Liver and CRC, with high sensitivity and specificity permitting them to propose that plasma eHsp90 can serve as pan-cancer biomarker [117].

Interestingly, like in the in-vitro studies, eHsp90 has also been found to associate with exosomes in clinical samples [118,119,120]. Peinado et al. identified a molecular signature on exosomes derived from patient samples, which included Hsp90 along with other proteins [119]. Hsp90 was specifically associated with exosomes in approximately 70% of metastatic melanoma patient samples but not in samples from healthy individuals or those with benign melanoma [119]. Similarly, EVs isolated from prostate cancer patients’ plasma samples contained elevated levels of Hsp90, which could distinguish between healthy donors and prostate cancer patients with high sensitivity and precision [120]. Hsp90-enriched EVs have also proven to be reliable predictors of metastasis in head and neck cancer and oral cancer [121]. These studies collectively highlight the significance of free and exosomal Hsp90 in invasive cancer and metastasis, establishing eHsp90 as a valuable biomarker for metastatic cancer and, in some cases, early cancer stages. These studies indicate that eHsp90 possess the potential to be a clinical biomarker for various cancers, even a pan-biomarker, and might even be a distinguishable factor for invasive or metastatic cancers against non-malignant primary tumor.

## 6. Conclusions and Future Directions

Intracellular Hsp90 overexpression and addiction is observed in many cancers and is extensively implicated in driving cancer progression and metastasis. Despite Hsp90’s role in these processes, clinical trials for pan Hsp90 inhibitors to treat cancer unfortunately were not successful, primarily due to Hsp90’s essential cellular functions such as homeostasis etc. leading to massive side-effects. Cancer cell secreted eHsp90 on the other hand is a promising therapeutic target, given its involvement and secretion in minimal physiological processes like stress response and wound healing. In cancer, eHsp90 plays a significant role in shaping the TME, particularly by influencing the ECM and ECM-regulating proteases to facilitate tumor invasion and metastasis. Reducing metastasis vs. tumor growth is a formidable clinical challenge but important to address given that most cancer-related deaths are due to metastasis and not from the primary tumor. In this context, targeting eHsp90 represents an excellent therapeutic target against various events of metastasis, especially invasion.

Understanding how eHsp90 acts in invasion and metastasis will provide insight to how best to target its cancer promoting functions. Our discussion has highlighted eHsp90’s involvement in various TME processes that are crucial contributors to tumor invasiveness, including ECM protease activity regulation, activation of CAFs, deposition and matrix regulation of FN and a recently identified crucial proinvasive function of Collagen-1 alignment. While not discussed here, eHsp90 impacts other premetastatic processes as well, such as angiogenesis, lymphangiogenesis, stromal cell migration and maintenance of cancer stem cells [39,43,122].

Unravelling the role of eHsp90 in driving invasion has paved the way for further exploration of its role in other metastatic steps and other pro-invasive processes to gain a comprehensive understanding of eHsp90’s broad impact as a master regulator of these processes. For instance, while existing studies have delved into eHsp90’s impact on individual ECM proteins and ECM regulating proteins, the broader effect of eHsp90 on ECM characteristics within the TME, including alterations in density, structure, and mechanical properties etc. remain unexplored.

So far, the studies have focused on conditioning of the TME during invasion close to the primary tumor it would be interesting to explore eHsp90’s role in for other metastatic events such as homing to secondary tissues and micro-metastasis at these locations. Particularly, parallels between processes at the primary invasive niche and pre-metastatic niches (PMNs) hint at the potential involvement of eHsp90 in PMN formation [123,124]. Given that eHsp90 orchestrates key processes at the primary site, including FN deposition, ECM remodeling, and CAF degradation, it is plausible that it plays a similar role in PMN creation.

Moreover, the known functions of eHsp90 could be explored beyond invasion and metastasis, for instance in enhance the effectiveness of existing therapies, such as CAR-T cell therapy, particularly in overcoming barriers posed by the TME. One of the significant challenges for CAR-T and other cellular therapies is the stiffened stroma, which can impede the ability of therapeutic cells to penetrate and reach tumor sites. Discussed in [125]. As mentioned in sections above, the stromal stiffness is driven by excessive deposition and crosslinking of ECM fibers—processes influenced by eHsp90 activity within the TME. [78,126]. Therefore, targeting and inhibiting eHsp90 may reduce ECM stiffness, facilitating better infiltration of CAR-T cells and other future cellular therapies into the TME, ultimately enhancing their therapeutic potency of various solid tumor malignancies.

Within the review we have mentioned the cell-impermeable inhibitors and antibodies developed to inhibit the activity of eHsp90 and its impact on invasion and metastasis. A detailed review of the eHsp90 inhibitors and antibodies most used has been summarized by Reynolds and Blagg [127]. It is that of note that current eHsp90 inhibitors only target ATPase activity in the N-domain [127]. While ATPase activity is key to many Hsp90 functions, other domains, such as the C-domain—crucial for dimerization—also play essential roles in eHsp90’s proinvasive actions, like Collagen-1 binding and alignment [32]. Developing cell-impermeable inhibitors targeting these additional domains could advance both our understanding of eHsp90’s full functional scope and the development of targeted therapies.

## Figures and Tables

**Figure 1 cancers-16-03873-f001:**
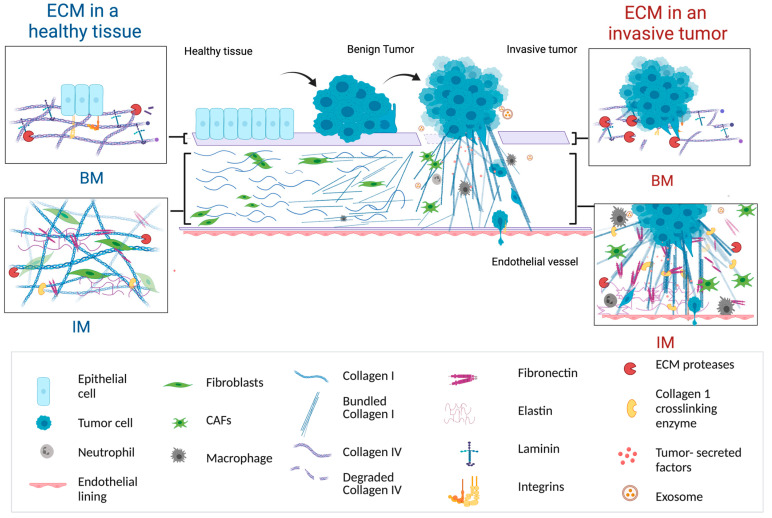
Depiction of BM and IM composition and structure in a healthy tissue microenvironment vs. invasive TME [51].

**Table 1 cancers-16-03873-t001:** Pro-metastatic role of eHsp90 in different Cancers based on in-vitro studies: including information regarding binding proteins, eHsp90-intervention molecules, and mechanistic insights.

Type of Cancer	ECM or ECM Proteases Involved	Agents Used to Study the Role of eHsp90	Mechanism	Cell Lines	Phenotype Associated w/Metastasis
Breast Cancer	MMP-2 [18,21,24,31,43]	mAb-4C5 [21], mAb [31,63], pAb [42], Geldanamycin on beads [18], 17AAG [32], DMAG-N-oxide [32], Novobiocin [32]	MMP-2 activation and stabilization [18,21,31]	MB231 [18,24] MB453 [21] SKBR3 [32] SUM1315 [42]	Matrigel invasion and migration [18,21] Endothelial transmigration and Angiogenesis [43]
	MMP-9 [21]	mAb-4C5 [21]	MMP-9 activation [21]	MB453 [21]	Not studied
	Plasminogen [20]	DMAG-N-Oxide [20]	Plasminogen activation [20]	T24 [20]	Cell migration [20]
	FN [25,44,46]	Novobiocin, Coumermycin [25,44,46]	Inhibition of FN internalization and FN matrix formation [25,44,46]	A172, MB231, MCF7 [25,44,46]	Not studied
	LOXL2 [27]	STA-12-7191 [27]	Binding to LOXL2 [27]	MB231 [27]	Cell migration [27]
	Collagen-1 [26]	STA-12-7191 [26]	Collagen-1 binding and fiber alignment [26]	MB231, SUM159 [26]	Collagen-1 Invasion [26]
Fibrosarcoma	MMP-2 [18,45]	DMAG-N-Oxide [18,45]	Activation and active MMP-2 stabilization [18,45]	HT1080 [18]	Invasion [18]
	LOXL2 [27]	STA-12-7191 [27]	Binding to LOXL2 [27]	A172, HT-1080 [27]	Wound Healing and Migration [27]
Melanoma	MMP-2 [21]	mAb-4C5 [21,33]	MMP-2 activation [21,33]	B16/F1 [21,33]	Endothelial transmigration and angiogenesis [21]
	-	DMAG-N-Oxide [32]	inhibition of eHsp90 activity [32]	B16F10 [32]	Matrigel invasion and wound healing [32]
Prostate Cancer	-	DMAG-N-Oxide [32]	eHsp90 activity inhibition [32]	PC3 [32]	Matrigel invasion and wound healing [32]
Bladder Cancer	-	DMAG-N-Oxide [32]	eHsp90 activity inhibition [32]	T24 [32]	Matrigel invasion and wound healing [32]
GBM	-	DMAG-N-Oxide [23]	Inhibition of LRP1 signaling [23]	U251, U87 [23]	Cancer cell migration and Matrigel invasion [23]
CRC	-	Ab [35]	Inhibition of LRP1 signaling [35]	HCT-8 [35]	Matrigel invasion [35]

**Table 2 cancers-16-03873-t002:** The Role of eHsp90 in Different Cancers based on in-vivo studies: Phenotype and mechanism associated.

Cancer Type	Agents Used to Study the Role of eHsp90	Cell Lines Used	Mouse Model Utilized	Phenotype	Mechanism Investigated
Breast Cancer	DMAG-N-Oxide [36]	MB468 [36]	SCID mice—Xenograft cancer model [36]	Lung and Liver metastasis [36]	eHsp90 secretion [36]
	HS-27 [38]	MB468 [38]	SCID mice—Xenograft cancer model [38]	Aggressive tumor phenotype [38]	Increased eHsp90 abundance and internalization by aggressive cancer [38]
	IG6-D7 [37]	MB231 [37]	Nu/Nu mice—orthotopic breast cancer model [37]	Tumor growth and metastasis [37]	eHsp90 inhibition and dual lysine motif inhibition reduces tumor metastasis [37]
	HCmAb2 [28]	MB231 [28]	Orthotopic breast cancer model [28]	None tested [28]	Binding to eHsp90 [28]
	Other mAbs [31,39]	MB231 [31,39] MCF-7 [39]	BLAB/c—orthotopic breast cancer mouse model [31,39]	Stromal and muscle local invasion; Liver and Lung metastasis [31] Lymph node metastasis and Lymphangiogenesis [39]	role of EEVD motif in eHsp90 secretion [31] Activation of LRP1 signaling pathway [39]
Lung Cancer	rHsp90α [40]	H2177 [40]	Orthotopic breast cancer model [40]	Local invasion and metastasis to Liver [40]	Role of Metformin in Hsp90 secretion [40]
Melanoma	DMAG-N-Oxide [32]	B16 [32]	Nu/Nu mice—Tail-Vein Injection [32]	Metastasis to Lungs [32]	Inhibition of eHsp90 activity [32]
	mAb-4C5 [30]	B16 F10 [33]	C57BL/6 [33]	Metastasis to Lungs [33]	Inhibition of eHsp90 activity [33]

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
