# Peer review of "The Role of eHsp90 in Extracellular Matrix Remodeling, Tumor Invasiveness, and Metastasis"

_cancers, 2024, doi:10.3390/cancers16223873_

Round 1
Reviewer 1 Report
Comments and Suggestions for Authors Title: The Role of eHsp90 in Extracellular Matrix Remodeling, Tumor Invasiveness, and Metastasis Authors: Pragya Singh, Daniel G. Jay COMMENTS: This is an interesting and comprehensive review which will be helpful for oncologists. The submitted manuscript is well written and nicely illustrated. I think that this material may be accepted in the present form, however, I would like to suggest a couple of minor points, discussions of which may strengthen the content of this review: 1. When the Authors discuss the approach of Tsutsumi et al. (Oncogene 2008), it seems important to emphasize that the cell-impermeant inhibitors of Hsp90 are expected to be much less toxic toward normal non-cancerous) cells as compared to cell-permeable inhibitors of Hsp90. (That is why such inhibitors may be more tolerable by patients and results of clinical trials may be better than those of cell-permeable inhibitors); 2. It would be nice if the Authors discuss targeting eHsp90 as a possibility to improve the potential of CAR-T cell therapy against solid tumors. (It is known that the dense ECM in solid tumors is an obstacle for migrating CAR--T cells; if it is possible to remodel the ECM in solid tumors by somehow targeting eHsp90, such an approach may help to spread the method of CAR-T cell therapy toward various malignancies.)Author Response
Question 1:
When the authors discuss the approach of Tsutsumi et al. (Oncogene 2008), it seems important to emphasize that cell-impermeant inhibitors of Hsp90 are expected to be much less toxic toward normal non-cancerous cells compared to cell-permeable inhibitors of Hsp90.
We appreciate this insightful suggestion. We have added a discussion to emphasize the comparative safety of cell-impermeant Hsp90 inhibitors for non-cancerous cells, potentially increasing their tolerability in clinical settings. This addition can be found in lines 386–394 of the revised manuscript.
Question 2:
It would be beneficial to discuss targeting eHsp90 as a potential method to enhance CAR-T cell therapy against solid tumors, considering that ECM density in solid tumors is a barrier to CAR-T cell migration.
Thank you for this excellent point. We have expanded the discussion to consider the potential of targeting eHsp90 to ECM and thus improve CAR-T cell therapy effectiveness in solid tumors. This addition can be found in the Conclusions and Future Directions section (second last paragraph), lines 501–510, and we believe it adds depth to the therapeutic possibilities discussed in this review.
Reviewer 2 Report
Comments and Suggestions for Authors
In this narrative review the authors discuss current literature on the role of extracellular Hsp90 in cancer invasiveness and metastasis. The authors have contributed papers in the field.
In this review the authors focused on the invasion process of tumor cells and in the activation of extracellular matrix proteases by the secreted Hsp90. Besides the molecular mechanism underlying the ECM remodelling the authors discussed the usefulness of eHsp90 as a biomarker in the clinical arena.
Author Response
In this narrative review, the authors discuss the current literature on the role of extracellular Hsp90 in cancer invasiveness and metastasis. The authors have contributed papers in the field.
In this review, the authors focused on the invasion process of tumor cells and the activation of extracellular matrix proteases by the secreted Hsp90. In addition to the molecular mechanism underlying the ECM remodeling, the authors discussed the usefulness of eHsp90 as a biomarker in the clinical arena.
Thank you for your positive feedback on our review. We appreciate your support and are glad to hear that the content aligns well with the current understanding in the field.
Reviewer 3 Report
Comments and Suggestions for Authors
Manuscript entitled "The Role of eHsp90 in Extracellular Matrix Remodeling, Tumor Invasiveness, and Metastasis"
Major issues:
1. Given that the Extracellular Matrix Remodeling is quite different in various cancer types. The authors are encouraged to include tables summarizing the validated roles of eHsp90 in vivo (one table) and in vitro (another table) and their targets in various cancer types.
2. The authors should provide more review on the efforts of developing therapeutic strategies targeting eHSP90.
3. The different biological roles of eHSP90 and HSP90 should be provided.
4. The rationale for producing eHSP90 and its upregulated mechanism should also be provided.
Author Response
Comment 1: Extracellular Matrix Remodeling varies significantly among cancer types. Consider adding tables summarizing the roles of eHsp90 in vivo (one table) and in vitro (another table) with their respective targets across cancer types.
Response 1: Thank you for this constructive suggestion. We have included two new tables—Table 1 and Table 2—that summarize in-vitro and in-vivo studies on eHsp90's role in invasion and metastasis (respectively) and the experimental approaches used to study these roles. These tables aim to provide a comprehensive overview of eHsp90 across various cancer contexts and enhance clarity.
Comment 2: The authors should provide a more comprehensive review of the development of therapeutic strategies targeting eHSP90.
Response 2: We appreciate your suggestion to expand on therapeutic strategies targeting eHsp90. While we recognize the importance of this topic, a detailed review of this subject is beyond the current scope of this manuscript. We aim to address this in future work, where we can focus on therapeutic advances more extensively.
Additionally, a recent review by Reynolds and Blagg summarizes cell-impermeable small molecule inhibitors that bind to the N-terminal domain of Hsp90 and antibodies targeting Hsp90α. [1] This review discusses the chemical properties of the N-domain inhibitors of eHsp90 and inhibitors of pan-Hsp90 that bind to other domains (M and C). [1] We have added a brief paragraph in the conclusion and future directions section discussing the current status of eHsp90 inhibitors, found in lines 511–520.
Comment 3: Provide further details on the differing biological roles of eHSP90 and HSP90.
Response 3: Thank you for this request. Several reviews have extensively covered the detailed biological roles of Hsp90 and eHsp90, referenced in our review. For instance - [2] [3] [4] [5] In response to your question, we have added information distinguishing the various homeostatic functions of Hsp90 compared to the minimal impact of eHsp90 in healthy cells. Specifically, lines 56–68 (of the manuscript) state that eHsp90 offers a potentially safer and more effective therapeutic strategy compared to pan-Hsp90 inhibitors, which affect intracellular Hsp90 and its many homeostatic functions such as correct protein folding, protein activation, ligand binding, and transcriptional regulation. [6], [7] Given Hsp90’s critical role in cell homeostasis and its interactions with approximately 200 proteins, inhibitors targeting the ATP-dependent chaperone function of intracellular Hsp90 have faced challenges in clinical trials due to high toxicity observed at effective doses, likely due to interference with Hsp90’s essential functions. [6], [7] [8]
Comment 4: The rationale for the production and upregulation of eHsp90 should be discussed.
Response 4: Thank you for an excellent question. While more studies are required to answer precisely why the eHsp90 production is upregulated and secreted in cancer, in my opinion, likely, Hsp90 protein reserves are already maintained in (2-3% of all intracellular) cells, so a quick secretion is possible upon experiencing stress such as hypoxia, wound or cancer. In contrast, more of Hsp90 is expressed concurrently. Also, Hsp90, a highly interactive protein, is likely to perform multifaceted functions concurrently and quickly via interaction with different proteins.
Since the exact reason is unknown, although the mechanism and regulation of eHsp90 secretion and activity have been studied, we have added a brief discussion highlighting eHsp90’s role as a multifunctional hub in cancer-promoting processes, including activating tumorigenic cell signaling, ECM degradation, ECM deposition, and matrix alignment. These updates can be found in the revised manuscript. Various mechanisms implicated in eHsp90’s secretion and activity, including phosphorylation, acetylation, and C-terminal cleavage, have been extensively studied and are included in reviews. [2] [9] [5] Further exploration of these mechanisms remains outside the current scope of this review.
Round 2
Reviewer 3 Report
Comments and Suggestions for Authors
The revision is acceptable.